# The Effect of Climate Parameters on Sheep Preferences for Outdoors or Indoors at Low Ambient Temperatures

**DOI:** 10.3390/ani10061029

**Published:** 2020-06-13

**Authors:** Peep Piirsalu, Tanel Kaart, Irje Nutt, Giovanni Marcone, David Arney

**Affiliations:** 1Chair of Animal Nutrition, Institute of Veterinary Medicine and Animal Science, Estonian University of Life Sciences, F.R. Kreutzwaldi 1, 51006 Tartu, Estonia; irje.nutt@emu.ee (I.N.); giovanni.marcone@student.emu.ee (G.M.); david.arney@emu.ee (D.A.); 2Chair of Animal Breeding and Biotechnology, Institute of Veterinary Medicine and Animal Science, Estonian University of Life Sciences, F.R. Kreutzwaldi 1, 51006 Tartu, Estonia; tanel.kaart@emu.ee

**Keywords:** animal welfare, cold climate, wind chill, sheep outdoors, animal shelter

## Abstract

**Simple Summary:**

Sheep may be kept indoors over the winter period, especially in cool climates. There is currently a drive to keep sheep outside, especially on organic sheep farms. This paper examines the preferences that sheep have in cool conditions for outdoor or indoor housing to inform management practices. We found no evidence that mature sheep should be kept confined indoors throughout the winter period, even in normal winter weather conditions in northern Europe, with temperatures as low as −20 °C and where precipitation and relative humidity may be high. In all conditions, during this trial, the majority of ewes preferred to be outside. Access to the outdoor area should be managed to restrict outdoor access for lambs, ewes with unweaned lambs and recently shorn sheep. Conditions in which sheep did choose to move indoors were: low wind chill values (≤10 °C) and/or high air humidity (>90%). In such cases, sheep should have the opportunity to shelter indoors.

**Abstract:**

Threshold temperatures for cold stress in sheep are not well understood, the available literature is somewhat dated and reports relate to winter temperatures that are relatively benign. Sheep’s preferences for outdoor versus indoor environments, when given free access to both, were investigated in the winter period at temperatures as low as −23 °C. Two sheep farms, one with access to a permanent uninsulated barn and one with a polytunnel shelter, both with free access to an outdoor area, were used. Observations were made with a camera positioned to register numbers of sheep outdoors and indoors, with one image taken hourly over twenty-four hours. The sheep clearly preferred to be outdoors; on all occasions the majority of the sheep were outdoors. There was, however, a significant decrease, albeit small, in the numbers of sheep choosing to be outdoors at lower temperatures (*p* < 0.001), higher relative humidity (*p* < 0.001) and greater wind chill (*p* < 0.001). Therefore, even at cooler temperatures than reported previously, sheep are motivated to be outdoors rather than indoors. It is not implicitly good for their welfare, and may not be true for lambs and shorn sheep, but accessing an outdoor area appears to be what they choose to do when given the choice.

## 1. Introduction

Recently, sheep farming has undergone changes in both conventional and in organic farms. While, in the past, sheep were often kept indoors in barns during the winter period in cooler climates, it is becoming more common to keep sheep outside in this period. While sheep are better adapted to cold temperatures than some other livestock, their lower critical temperature, which in ideal conditions may be as low as −20 °C, can vary widely between shorn and unshorn sheep, in wind or rain [1] and, presumably, also between breeds and between individuals of the same breed in the same management system under the same climatic conditions [2]. Unshorn and recently shorn sheep may choose different resting locations in free-ranging conditions [3]. During calm weather, and at night, sheep remain as one flock, but in windy weather most shorn sheep congregate in a shelter, while unshorn sheep remain away from shelter. The lower critical temperature depends on the wool length; for unshorn sheep with 10-mm-thick wool, the lower critical temperature has been found to be 25 °C, for sheep with 50-mm-thick wool it is −5 °C and for sheep with 70-mm-thick wool it is −18 °C [4]. Nevertheless, sheep have been assumed to be in a state of cold stress at an ambient temperature of +7 °C in wind and while wet [5].

How sheep experience cold and their preferences in different climatic conditions is unclear. It is important to consider animal preferences regarding the different possibilities offered to them, which would enable improved systems of animal husbandry [6]. The literature investigating the behavioural preferences of sheep comprises a limited number of studies related to the interplay of farm facilities: flooring [7], ambient temperature, feeding and vaccination [8]. Keeping sheep indoors protects them from the variation in weather and unpredictably of environmental conditions. However, housing sheep indoors over the winter can have poor consequences for sheep’s wellbeing, derived from decreased space allocation [9], a worse indoor climate than outdoors, or hygiene [10]. Caroprese [11] and Cassamassina et al. [12] concluded that the welfare of housed sheep depends on the possibility of free access to external areas, protection from thermal extremes [13], suitable flooring [7], sufficient space allocation [9], careful control of the indoor climate and hygiene [14], proper ventilation and light regimes [11]. In addition, outdoor ewes have reportedly lower milk somatic cell counts compared to indoor ewes [12].

Organic farming is becoming increasingly popular globally [15]. A tenet of organic animal husbandry is good animal welfare, which is achieved by providing conditions for animals which enable them to behave in the most natural manner possible [16]. Pursuant to the requirements of organic animal husbandry, sheep must have permanent access to walking areas, whenever the health of the animals, weather conditions and the state of the ground allow [17].

There is a lack of studies determining the behavioural preferences of sheep regarding being outside or inside in the winter period. It is especially important to know the behavioural preferences of animals in extreme conditions during this period (low temperatures, strong wind, precipitation), to help determine unsuitable conditions for animals and enable the planning of husbandry systems based on the preferences of animals.

The purpose of this study was to determine the behavioural preferences of ewes in regard to choosing to be indoors or outdoors in a walking area, when given free access to both, at cold temperatures during the winter period, and to identify unsuitable climatic conditions for keeping sheep outside during the wintertime in cool climates.

## 2. Materials and Methods

The study was carried out on two organic sheep farms (Farm A, Farm B) where the sheep were allowed access to an outdoor area all year. On both sheep farms, the animals were kept outside at pasture in the spring, summer and autumn. In winter, the sheep were kept in an outdoor paddock from where they were free to walk into an uninsulated barn (Farm A) or polytunnel (details of this can be found in the farm description below—Farm B). This type of sheep farming is common in Estonia.

### 2.1. Description of Farms

Farm A is located in Tartu County, Estonia (latitude: 58°34′44″ N and longitude: 26°44′51″ E), the flock is comprised of the Estonian Whiteface Sheep breed (flock numbering 260 ewes in 2017, mean age of 6 years and 170 ewes in 2018, mean age of 7 years). 

The grazing season constitutes approximately 260 days a year, during which the sheep are grazed on a rotational basis in five different paddocks. In December and January, when the outdoor yards were covered in snow, the sheep were additionally fed rolled grass hay in the paddocks. The sheep were brought into the barn on 15th January 2018 when the weather conditions worsened and were returned to pasture at the beginning of May. The lying area in the uninsulated barn was covered with straw and hay bedding and the size of the lying area was 240 m^2^. In 2018, the sheep spent approximately 105 days in or next to the barn, during which time they could freely choose whether to be in the outdoor area or in the barn. The outdoor area, with an area of 520 m^2^, was covered with straw and hay bedding. Sheep were shorn in August 2017 and 2018. The lambing period of ewes started on 1st April 2017 and on 5th March 2018 and lasted for about 30–40 days. The ewe and lamb were placed into an individual pen in the barn where they spent 1–2 days together before they were allowed to join the rest of the flock. During the winter period, sheep were fed with hay, silage, molasses, salt and a mixture of minerals. All forms of fodder were offered to the sheep ad libitum both in the barn and in the outdoor area. Freeze-proof drinkers for sheep were placed inside the barns.

Farm B is located in Viljandi County, Estonia (latitude: 58°14′26″ N and longitude: 25°42′31″ E) and they also keep the Estonian Whiteface Sheep breed. The husbandry used on this farm was generally similar to that in Farm A. The main difference lay in the use of a McGregor polytunnel (McGregor Group, Laceys Farm, Bramdean, Hampshire, UK), its measurements being 9 × 18 m, with a lying area of 162 m^2^. This comprises a frame with a white polythene roof sheet. The number of ewes kept in the polytunnels was 121, with a mean age of 6 years. As with farm A, the sheep had continual free access to an outdoor area. According to usual practice on this farm, the sheep were brought into the polytunnel in February, i.e., approximately one month before the expected lambing season, as the lambing period of 2018 started on 23rd March. Starting from November 2017, silage and hay were additionally provided to the sheep in the outdoor paddocks. On this farm, the sheep were kept outdoors for about 290 days, starting from the end of April until the middle of February. The sheep were kept in or next to the polytunnels for 75 days, which included the fifth month of pregnancy, lambing and the first month of lactation. The ewe and lamb were placed into an individual pen in the polytunnel where they spent 1–2 days together before they were free to join the rest of the flock. While the sheep were kept in the polytunnel, they were fed grass silage, grass hay, a salt lick and a mixture of minerals offered both in the walking area and in the polytunnel ad libitum. When the naturally available water was not frozen, the sheep drank water from a ditch. When the water was frozen, they ate snow. Ewes were shorn in June 2017 and in July 2018. The size of the outdoor area was approximately 900 m^2^. Hay from the upper layer of the hay roll was used as bedding in the walking area and in the polytunnel. At the end of the grazing season, the layer of bedding measured about 30 cm in depth.

### 2.2. Observations

Observations of ewes were carried out in the winter when the sheep had free access to both the outdoor walking area and the barn/polytunnel.

Observations started in 2017 only on Farm A. These began on 30th March and lasted until 26th April, including 28 observation days in total. In 2018, observations were carried out on both farms. In Farm A, observations were carried out from 16th January until May 4th, 91 days in total (18 days were excluded due technical problems); and in Farm B, from 6th March until 23rd April, 49 days in total.

In order to determine sheep preferences, battery-powered trail cameras LTL-6310 Acorn (in Farm A in 2017), Wild Guarder WG-890-3G (Farm A and B in 2018) were placed indoors so that all the sheep indoors were visible and were photographed once every hour during the survey period. We could therefore determine the number and proportion of sheep indoors and outdoors. A database was created, which documented the total number of ewes, the number and percentage of ewes in the walking area and the number and percentage of ewes in the shelter each hour.

In parallel with the above, meteorological data obtained from Jõgeva Meteorological Station (Farm A) and Viljandi and Massumõisa Meteorological Stations (Farm B) were added for each observation hour [18]. These included ambient temperature (degrees Celsius), wind direction (compass degrees), wind speed (m/s), maximum wind gusts (m/s), wind chill/perceived temperature, relative air humidity (%) and atmospheric pressure (hPa). Wind chill was calculated with the formula
$$T_{WCC}=13.12+0.6215\;\cdot \;T_{A}-13.947\;\cdot \;V_{10m}^{0.16}+0.4867\times T_{A}\times V_{10m}^{0.16}$$
where *T_WCC_* is average wind chill; *T_A_* is average air temperature per hour and *V*_10m_ is average wind speed per hour [19]. Additionally, data were gathered for cloud cover (1—Clear, 2—Overcast, 3—Partly cloudy; 4—Fog) and precipitation (1—Rain, 2—Snow, 3—Sleet, 4—Without Precipitation).

As a result of the survey, a database was created for Farm A, which contained 28 observation days for 2017 (from 30th March–26th April), during which there were a total of 646 observations. For 2018, the survey in Farm A contained data for 91 observation days (from 16th January–May 4th), during which 1888 observations were made. Due to technical issues, observations were not carried out in 2018 during the period of 17–25 January and 28 March–4 April. On Farm B, the observations were carried out for 49 days in 2018 (from 6th March–23rd April), during which 1144 observations were made. The total number of observations included in the database file was 3678 for the two farms over two years (Appendix A).

### 2.3. Statistical Analysis

All statistical analyses were performed and figures were constructed with R 3.5.1 [20]. As it was not possible to identify the ewes from the photographs, only total numbers of ewes inside were counted. On both farms, ewes from lambing pens were excluded on the day of lambing from the total number of ewes. From these data, the numbers of ewes outside and percentages of ewes inside and outside were calculated. The last were used to study the associations with climate parameters.

The relationships between the percentage of ewes observed outdoors and meteorological data were studied with a linear correlation analysis. To identify the potential non-linear relationships, locally weighted scatterplot smoothing (LOWESS) was used within R. Additionally, the observations were divided into four groups, ≤70%, 71–80%, 81–90% and 91–100% of the ewes outside, and the means of climate parameters corresponding to these groups of observations were compared by an analysis of variance followed by Tukey’s post-hoc test (R functions glht and cld of package multcomp [21]). The distributions of climate parameters with the percentage of ewes outside were visualized with beanplots (R function beanplot of the package with the same name). All results were considered statistically significant at *p* ≤ 0.05.

We also applied multiple regression, random forest and several other machine learning algorithms to discover common patterns in climate and time parameters predicting the ewes’ behaviour. However, as with time period and geographical region of the study, the date and time within day are related to climate characteristics and the latter, in turn, are related to each other, the results of the relative significance of the effects and uniqueness were inconsistent and depended more on the pattern discovery algorithm than on the real dependencies. Due to this, only the more robust results of simple analyses are presented here.

## 3. Results

### 3.1. Summary of Observations

The weather conditions were variable in the study period. The average ambient temperature over all observational hours and farms was 0.2 °C (standard deviation 7.7 °C, minimum −23.3 °C and maximum 21.8 °C); being the lowest in February and March with mean values of −9.7 °C and −3.2 °C, respectively. The mean wind speed was 2.5 m/s (standard deviation 1.5 m/s, minimum 0.0 m/s and maximum 12.4 m/s) and the mean relative air humidity was 78.6% (standard deviation 18.5%, minimum 9.0% and maximum 100.0%).

Observations showed that ewes generally preferred to be outside and spent considerably less time in the barn/polytunnel during the study period (Figure 1). On average, throughout the period, 92.3% (95%CI = (91.9, 92.6)) of ewes in Farm A and 93.3% (95%CI = (92.9, 93.7)) of ewes in Farm B were outside. The minimum percentage of ewes outside during the observation period was 50.6% in Farm A and 62.0% in Farm B. In Farm A, there were 14 occasions with 60% or fewer ewes outside and, at all these moments (except one), the ambient temperature was less than −15 °C and the wind chill was less than −20 °C. On Farm B, there were no observations at such cold temperatures, but in all three observations with less than 65% of ewes outside, both the wind chill value was below zero and the relative air humidity was over 90%. On only 8.6% of the total number of observations were more than 20% of ewes observed to be indoors. Moreover, only slightly more than a quarter of the cases (27.5% of the observations) were more than 10% of the ewes indoors. On the other hand, 12.5% of observations showed that no ewes were indoors (i.e., all ewes preferred to be outside in the walking area).

Ewes spent slightly less time outdoors at night on both farms (from midnight until 6.00–7.00 a.m.) compared to the daytime (from 9 a.m. until 6.00–7.00 p.m.). On Farms A and B, respectively, 88 and 91% of ewes chose to stay outdoors at night. From 13.00 to 18.00 on Farm A and 9.00 to 22.00 on Farm B there were always more than 70% of ewes choosing to be outdoors (Figure 2).

### 3.2. Preferences of Ewes Depending on Climatic Conditions

Regardless of the ambient temperature on both farms, the majority of ewes chose to stay outdoors (Appendix A). Nevertheless, a significant positive correlation (albeit weak) was observed between the ambient temperature and the percentage of ewes observed outdoors (on both farms r = 0.23, *p* < 0.001); the lower the temperature, the more ewes chose to stay indoors.

The correlation was almost the same in the case of wind chill where, in addition to ambient temperatures, wind speed was also taken into account (Figure 3). When the wind chill temperature was between −5 °C and −10 °C, the mean percentage of ewes outside was 92–93%, whereas when the wind chill temperature fell below −10 °C, on average less than 90% of the ewes remained outdoors. When the wind chill temperature fell below −20 °C (which only happened on Farm A), less than 85% of the ewes were outside. Nevertheless, even in such extreme cold weather, the percentage of ewes outside did not fall below 50% in any of the observations in these conditions.

Relative air humidity had a negative correlation with the number of ewes outside, and more ewes were observed indoors at a relative air humidity of above 80–90% (Figure 4) than at a lower level of humidity. However, even with relative air humidity at 100%, only a few observations found more than 30% of the ewes indoors. When the relative air humidity was lower, the ewes preferred to be outdoors, with the exception of cases when low air humidity was accompanied by an anticyclone and very low temperatures in winter, as was the case in Farm A.

Wind speed alone was not significantly related to indoor/outdoor choices. The relationship between atmospheric pressure and the location of ewes was unclear and was more likely to reflect the weather conditions (ambient temperature, wind, precipitation). The grouping of climate conditions according to cloud cover and precipitation did not add anything—the average proportions of ewes staying outside varied only by 1–2% in different groups without clear causes.

Grouping meteorological data gave similar patterns of indoor/outdoor choices by the sheep. When at least 30% of the ewes were observed to be indoors, both the mean ambient and wind chill temperatures were significantly lower compared with the temperatures when fewer ewes were observed indoors (Table 1). However, this difference was only apparent in Farm A since the observation period included the coldest month of the year, February. In addition, significantly higher wind chill temperatures were recorded when less than 10% of ewes were observed to be indoors. Nevertheless, relatively high percentages of sheep chose to be outdoors at a range of wind chill temperatures (Appendix A). The relative air humidity was the lowest when less than 10% of ewes were observed indoors, and in Farm A observations with the highest number of ewes in the shelter also had the highest wind speeds (Table 1).

The analysis of the combined effect of ambient temperature, wind speed and air humidity on the location of the animals showed that the most unfavourable conditions were low wind chill values with high relative air humidity. On both farms, there were fewer ewes outside when the average wind chill was lower and the average air humidity was higher. Moreover, if the air humidity was below 60% and, simultaneously, the wind chill was over 0 °C, there were almost no observations with less than 80% of ewes outside. By omitting only the most extreme weather conditions with wind chill values below −25 °C, at all wind chill and air humidity values, more than 90% of ewes were observed to be outside (Appendix A).

The analysis of wind direction indicated that, on both farms, on average, 5–8% more ewes preferred to be in the shelter when the outdoor area was exposed to the wind, compared to when trees, buildings or topography provided protection from the wind direction (Appendix A).

## 4. Discussion

It is clear that, when given the choice, ewes take advantage of the opportunity to be outdoors; even when temperatures are well below freezing, over half of the ewes chose to be outdoors even at temperatures as low as −20 °C. Nevertheless, there was a significant correlation between air temperature and the number of ewes choosing to be outdoors in both the housing and polytunnel systems. Taking wind chill into account, as temperatures dropped more, ewes moved indoors, but this was never more than 50% of the flock. There was a similar pattern for humidity, with more ewes choosing to be indoors at higher relative humidity, but there was never more than 50% of ewes indoors, even at 100% relative humidity. Wind speed alone, cloud cover and precipitation had no significant effects on the indoor/outdoor choices. However, wind direction was important; if the outdoor had no shelter from the wind direction, this resulted in fewer animals (5–8%) remaining outside (Appendix A).

Diurnally, the ewes were more likely to be indoors during the night (from midnight to 07.00 a.m. in particular) than during the day (from 09.00 to 19.00). However, even at night the majority of the ewes were observed outdoors rather than indoors. These findings were consistent across both farms.

There was therefore no evidence that the sheep avoided the outside areas at the range of temperatures in this study (minimum −23.3 °C and maximum 21.8 °C). These findings only illustrate the preferences of sheep in these weather conditions and it is not therefore conclusive that welfare is necessarily improved. There may be stressors involved with being outside, including health problems arising from muddy and wet conditions underfoot, and the increased risk of predation. Indeed, during the period of this study, part of one of the observed flocks (of breeding rams) fell victim to a wolf attack, in paddocks close to the homestead. Eight animals were killed as a result. Thus, being outside is not necessarily good for sheep welfare. Nevertheless, this study does show that, when given a choice, sheep take advantage of the opportunity to be outside even in what might be expected to be adverse weather conditions, potentially leading to cold stress. Temperatures as low as −20 °C did not appear to be aversive to the sheep, as most of the animals preferred to be outdoors even at this temperature. Considering shorn sheep, shearing took place on Farm A in August, and on Farm B in June (2017) and August (2018), so there would have been around four-six months of wool growth. This seems to be a sufficient amount of fleece for outdoor sheep in winter. This is in agreement with Ekesbo [4], who reported that sheep’s lower critical temperature might be −18 °C when they have a wool staple length of 7 cm.

The timing of lambing is important. Preferred lambing periods in northern Europe are in March and April if sheep are kept outdoors in winter, but even in these months air temperatures can be relatively low, certainly below freezing. Barns or polytunnels are suitable for lambing pens where they are protected against rain, snow and wind. Free access to the outdoors may not be suitable during this period, as the ewes may choose to go outside where the lambs will follow them [22], thus exposing the lambs to risk of hypothermia. The temperatures recorded in this study are far below those suggested as the lower critical temperature for newborn lambs of an ambient temperature of 27 °C, and lambs shiver at 10 °C and less [23].

It may be argued that behavioural synchronicity is a factor here, with a small number of leader sheep determining the behavioural choices of the flock as a whole [24], but it is the authors’ contention that the pattern of choices on the multiple dates and weather conditions and on both farms make this an unlikely dominant cause in affecting the observations recorded. Further work on this, with different ages, breeds and sexes of sheep and the tracking of individually identified sheep (the absence of this was the main limiting factor in the present study) would allow for a study of the overall consistency of sheep behaviour. In addition, considering the potential effect of time to and from lambing and fitting more complex statistical models could add to the better understanding of the preferences of sheep to be either indoors or outdoors in cold conditions. In terms of climate parameters, more data from a larger number of farms may allow for the elucidation of more complex relationship patterns and to study the effect of random and extreme fluctuations in climates (which are now potentially more frequent due to climate change).

## 5. Conclusions

The majority of ewes observed preferred to be outside regardless of the weather conditions during winter in a cool climate, at temperatures as low as −20 °C. At night, there was a small proportion of ewes that chose to move inside. The particular weather conditions that induced some of the sheep to move inside were low wind chill values (≤10 °C) and/or high air humidity (>90%).

Where the outdoor area was exposed to the wind, some of the ewes preferred to be in the shelter. It is therefore concluded that, in order to promote the welfare of sheep, they should be given access to an outdoor enclosure, even in cold and wet conditions. However they should also have the possibility of shelter and the option to access an indoor area, particularly at night. This is not to say that it is inherently better for the welfare of the sheep to be outdoors at these temperatures, and shorn sheep and lambs may well have shown quite different locational choices. However, it is clear that, when given the choice, the local Estonian breeds (at least) choose to be outside at cool temperatures typical of winter in a cool climate.

## Figures and Tables

**Figure 1 animals-10-01029-f001:**
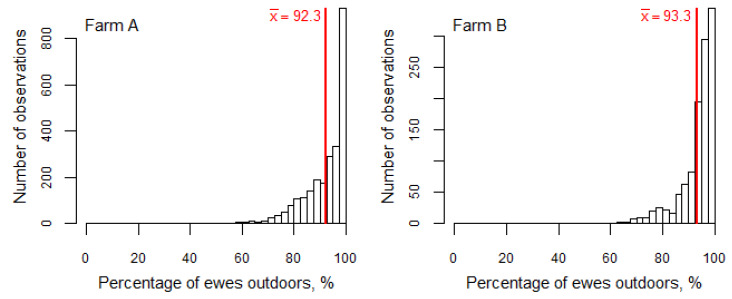
Percentage of ewes outdoors over the whole study period, red line and numerical value indicate the mean percentage.

**Figure 2 animals-10-01029-f002:**
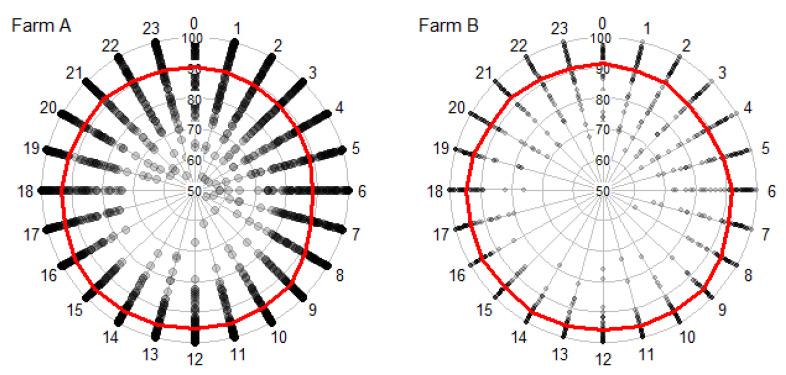
Percentage of ewes outdoors dependent on time of day. One point corresponds to one photograph and the red line indicates the mean percentage of ewes outside.

**Figure 3 animals-10-01029-f003:**
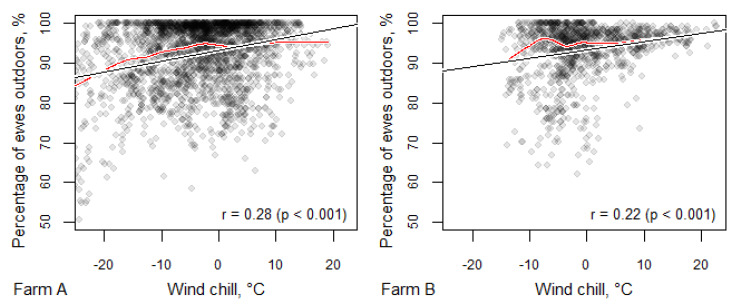
Percentage of ewes outdoors dependent on wind chill. One point corresponds to one photograph, black lines denote the linear relationship (corresponding correlation coefficients with *p*-values are presented in the lower right corners of the figures) and the red lines indicates the potential non-linear relationship fitted with a locally weighted scatterplot smoothing (LOWESS) curve.

**Figure 4 animals-10-01029-f004:**
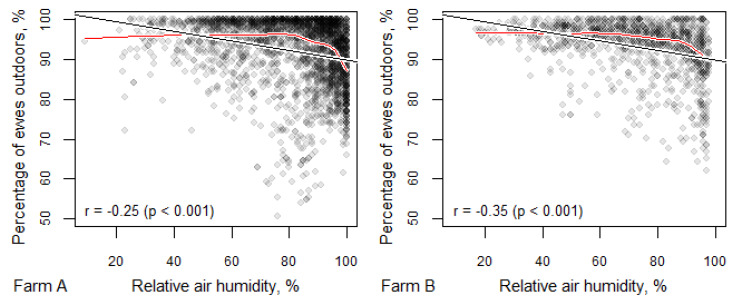
Percentage of ewes outdoors dependent on relative air humidity. One point corresponds to one photograph, black lines denote the linear relationship (corresponding correlation coefficients with *p*-values are presented in the lower left corners of the figures) and the red lines indicate the potential non-linear relationship fitted with a locally weighted scatterplot smoothing (LOWESS) curve.

**Table 1 animals-10-01029-t001:** Number and percentage of observations and average climate parameters (with standard errors in brackets) depending on the percentage of ewes outdoors in Farms A and B. Means without a common superscript letter in the same rows are significantly different (*p* < 0.05, Tukey’s post-hoc test).

Meteorological Data	Farm	Percentage of Ewes Outdoors			
		≤70%	71–80%	81–90%	91–100%
Number of shots (percentage)	Farm A	53 (2.1%)	189 (7.5%)	558 (22.0%)	1734 (68.4%)
	Farm B	13 (1.1%)	65 (5.7%)	149 (13.0%)	917 (80.2%)
Ambient temperature, °C	Farm A	−13.4 (1.15) ^a^	−1.3 (0.58) ^b^	−2.2 (0.36) ^b^	0.6 (0.18) ^c^
	Farm B	−1.9 (1.32) ^a^	−1.3 (0.47) ^a^	−1.4 (0.42) ^a^	2.6 (0.21) ^b^
Wind chill, °C	Farm A	−19.8 (1.32) ^a^	−5.0 (0.68) ^b^	−5.8 (0.38) ^b^	−2.4 (0.19) ^c^
	Farm B	−4.2 (1.36) ^ab^	−4.2 (0.48) ^b^	−4.1 (0.45) ^b^	0.4 (0.24) ^a^
Wind force, m/s	Farm A	3.37 (0.15) ^a^	2.67 (0.11) ^b^	2.49 (0.07) ^b^	2.54 (0.04) ^b^
	Farm B	1.84 (0.20)	2.41 (0.12)	2.60 (0.13)	2.43 (0.05)
Relative air humidity, %	Farm A	84.0 (1.00) ^ab^	90.6 (0.90) ^c^	87.6 (0.64) ^bc^	78.2 (0.43) ^a^
	Farm B	92.5 (1.63) ^a^	85.6 (1.75) ^a^	83.9 (1.19) ^a^	68.6 (0.64) ^b^

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
