# Peer review of "The Effect of Climate Parameters on Sheep Preferences for Outdoors or Indoors at Low Ambient Temperatures"

_animals, 2020, doi:10.3390/ani10061029_

Round 1

Reviewer 1 Report

Two sheep farms, one with access to a permanent uninsulated building barn and one with a polytunnel shelter, both with free access to an outdoor area, were used.

Could use less commas and be easier to read as:

Two sheep farms, both with free access to an outdoor area, one with access to a permanent uninsulated building barn and one with a polytunnel shelter.

... what they choose to do when given the choice to do so.

Would use less words as:

... what they choose to do when given the choice.

This sentence has been separated from the one before it but it now lacks what it is about.

Presumably also between breeds and also between individuals of the same breed in the same management system under the same climatic conditions [2].

IT could be clearer

Presumably there is also variation between breeds and also between individuals of the same breed in the same management system under the same climatic conditions [2].

Further work on this, differences in different ages, breeds …

Would be clearer as:

Further work on this, in different ages, breeds …

Author Response

Dear reviewer,

Thank you for your revisions. We tried to correct our manuscript on the base of your comments. Please find those in a separate file.

Thank you for your suggestions,

Authors 

Reviewer 2 Report

The paper addresses an interesting topic in sheep welfare. Indeed, nowadays it is still difficult to estimate the sheep tolerance of cold temperature. Although the authors did not use physiological parameters of stress, they collected and analysed several behavioural data to analyse sheep preferences in being either out- or inside. It would be interesting to read more about the farming conditions and the features of the local breed involved. Still, the discussion could be expanded referring to other variables that could affect thermal comfort-related behaviours.

Major comments:

It is unclear if the farms were composed only by ewes during the observational period. The co-presence of lambs could have influenced the behaviour of the dams. If only ewes were present, it would be helpful to specify the age of the subjects (or the average of the flock) as it could influence the perception of the temperature. Some confusion derives also from the discussion of lambing related to the temperatures (LL 292-299). Furthermore, it could be preferable to support the results pointing out more the statistic behind them (see for instance the comparison at LL 194-197).

Minor comments:

L 84: On both sheep farms, the sheep animals were kept outside

L 126: please change with “Observations”

L 143: Wind chill factor is in some case argument of debate. Please cite a reference to support the formula used

L 148: number 128 should be 28

L 157: please to cite R use “R Core Team (2019). R: A language and environment for statistical computing. R Foundation for Statistical Computing, Vienna, Austria. URL https://www.R-project.org/”

L 167: please, cite the authors of “multcomp” package: Torsten Hothorn, Frank Bretz and Peter Westfall (2008). Simultaneous Inference in General Parametric Models. Biometrical Journal 50(3), 346--363.

L 179: although the statistical models are explained in the previous paragraph, it would be better to specify the exact value of p, standard error and interval of confidence

L 197: Farm B there were no days with fewer than 70% 70%

L 203: (Figure S1) I think you are referring to figure 2

LL 284-287: please, rephrase this sentence. Maybe it would be better to avoid the term “stress” as you did not measure it (as you say). Furthermore, the discussion on pro and cons in being outside includes more topic other than predation. Finally, the huddling behaviour of sheep is not clear in the context.

L 289: the hair (wool) length was not measured, so how do you know about this measure? In the literature the studies on grow rate of hair in specific breed. If you were referring to one study please refer to it.

Author Response

(The authors gave the same response as above.)

Round 2

Reviewer 2 Report

Dear authors,

Thanks for your reply and for addressing some questions. I have read the new manuscript without track changes, so reference to lines could be a little different as I am understanding that you referred to the file with printed track changes.

I will try to clarify my doubts:

  • About lambing in farm A, year 2017:
    • Lambing started 1st of April and lasted 30-40 days (L 101), where sheep had been maintained for 1-2 days in individual pens (L 102). The observational period started on the 30th of March. How did you exclude these sheep from your analysis? These sheep had no free choice to go out. If this was the case the percentage of sheep outdoor would change a little bit.
  • About lambing in farm A, year 2018
    • Lambing started 5th of March and lasted 30-40 days (L 101), where sheep had been maintained for 1-2 days in individual pens (L 102). The observational period started on January 16th and finished on May 4th. How many sheep were pregnant? Do you think that being pregnant had some chance to alter their behaviour? Please add some comments on this in the discussion.
  • About lambing in farm B, year 2018
    • Sheep were brought to polytunnel in February, lambing started 23rd of March and then the sheep were kept in or next to the polytunnels for 75 days, which included the fifth month of pregnancy, lambing and the first month of lactation (LL 116-118). Sheep had been maintained for 1-2 days in individual pens (L119). The observational period occurred from 6th of March to April 23rd. How did you exclude the sheep from your analysis? These sheep kept in individual pens had no free choice to go out. If this was the case the percentage of sheep outdoor would change a little bit.
  • Discussion about lambing (LL 292-299) is unclear by the fact that it was no part of the study and so, even if it reasonable to keep lamb in a warm environment, there were no data about lambs’ clinical conditions (e.g. temperature), so it seems to be out of context to infer about the risk of hypothermia. It would be helpful to see that you did not record any particular signs in lambs. Still, you could refer to some papers on this topic.
  • LL 143-147: you specified the formula used for Wind chill equivalent temperature index, but it is unclear why you adapted the formula (I am referring to factors: “-13.956” and “+0.4867”). In some case the wind chill is equalled to the air temperature at wind speeds of less than or equal to 1 km h–1, was this the case?
  • L 179: although the statistical models are explained in the previous paragraph, it would be better to specify the exact value of p, standard error and interval of confidence
    • Response: In the first chapter of results the summary/descriptive statistics of observations is given only. This means that only observed values in both farms are presented and no estimation and comparison are done. Due to this, it is nonsense to present any accuracy characteristics. In the following chapters where the relationships with weather conditions are studied, also p-values are presented.
    • Reply: When you say “ewes generally preferred to be outside and spent considerably less time in the barn/polytunnel during the study period” you do a comparison. Without statistical support the means have to be considered equals. This to say that words as less or more are meaningless in this scientific context.
  • Still, the sheep have been divided into four groups depending on the percentage of sheep outdoor and compared for the climate parameters. Was is the time of the day considered in the analysis? You found some differences, I am wondering if these could be related to the routine of farm management.
  • The discussion is in general based on descriptive statistic and therefore inference needs to be cautious. In the linear correlation test, the results indicated very weak correlations and coefficient of determinations thus are very smalls. This last parameter can be more useful, especially with large data sets where even week correlation can be significant.

All in all, I think that the data collected so far are very interesting and valuable but they should be presented with fewer insinuations.

Author Response

Dear reviewer,

Thank you for your revisions. We have corrected our manuscript on the basis of your comments and responded to your comments. Please see the attachment.

May, 4, 2020

Thank you very much for your efforts to improve the article,

Authors

This manuscript is a resubmission of an earlier submission. The following is a list of the peer review reports and author responses from that submission.

Round 1

Reviewer 1 Report

I found this a very interesting study and would like to see the authors make further progress in this area with respect to animal choice.

The time of day and location in which the animals were fed will have a strong influence on animal behaviour has been neglected in the materials and methods. It may be that the indoor and outdoor feeding facility were filled at different times and that lured the animals back and forth. It would also be excellent if feeding time could be displayed on the graphics in Figure 2. I would liek to say that these graphics are very good descriptors of the data, and perhaps with advice from the journal could be made even better to help readers take in the records at a glance – though I am unsure of the ability to display colour in the journal. The graphics in Figure 2 might also be improved by adding a shaded patch or simple lines to show the approximate hours of darkness and feeding times.

I remain a little confused by the discussion around the “night” (midnight to 7 am) compared with the “day” (9 am to 7 pm) at lines 283 to 284. This is 7 hours of night and 10 hours of day during the winter when nights are longer than days?

The authors refer to a “cold barn”, which seems from the text to be an uninsulated barn which is cold, rather than a “cold barn” which suggests it is kept cold, I think it is safe to simply call it a “barn”

Farm B came into the project later and was studied for a shorter period of time in only one year. As I understand the methods at lines 153 to 161, the observations began on January 16th but then a technical issue arose and data were not collected from the 17th until the 25th of January. Why not start the period on the 25th of january and just have the one technical issue from March 28 to April 4? Only one day of data lost and the analysis would be more concise.

18 management practices of sheep farming in cool conditions. There is no supporting evidence from

18 management practices. There is no supporting evidence

29 not well understood, and the available from

29 not well understood, the available

30 somewhat dated and at winter temperatures

30 somewhat dated and reports are from winter temperatures

45 It is not clear why the authors refer to “conventional and organic sheep farms” at this point in the manuscript. This journal is about animals and this issue about welfare, while the conventional or organic systems are market preference. The topic of organic farming is mentioned again at line 76, which is perhaps more appropriate, but this reviewer is not convinced that being able to behave in a natural manner is a tenet of organic farming. Conventional farms may house their animals indoors in the belief that housing improves welfare. Perhaps the authors are trying to understand what the market wants which could be more directly stated without labelling this organic/conventional/intensive.

48 other livestock animals, their lower critical temperature

48 other livestock, their lower critical temperature

Also at line 48 (which in ideal conditions may be as low as -20 °C) should be removed from parentheses and included in the sentence or in another sentence. The sentence that begins on line 47 is long and cumbersome.

54 depends on the wool length of sheep;

54 depends on wool length;

58 In addition to cold as a stressor and consequences for welfare, consequences of cold include reduced feed intakes and impairment of body condition through the need to generate heat internally [6].

58 In addition to acting as a stressor, cold can reduce feed intake and impair body condition through the need to generate heat internally, which can also have welfare consequences [6].

62 unclear, and from even a cursory reading of the above paragraph, the

62 unclear, and from the above paragraph, the

63 And it is important

63 It is important

72 depends on the possibility of free access

72 depends on access

74 “outdoor ewes” should be described as “ewes held outdoors” and contrasted with “ewes housed indoors” rather than “indoor ewes”

79 whenever the health condition of the animals, weather conditions and the state of the ground allow this [18].

79 whenever the health of the animals, weather conditions and the state of the ground allow [18].

83 period (low temperatures, strong wind, precipitation), which would help determine

83 period (low temperatures, strong wind, precipitation), to help determine

84 unsuitable conditions for animals and would enable the planning

84 unsuitable conditions for animals and enable the planning

94 In winter, the sheep were kept all of the time in a walking area paddock from where they were free to move into an uninsulated cold barn (farm A) or polytunnel (farm B). Such a manner of sheep farming is common in conventional as well as organic sheep farms in Estonia.

94 In winter, the sheep were kept in an area from where they were free to walk into an uninsulated barn (farm A) or polytunnel (farm B). Such a manner of sheep farming is common in Estonia.

101 on a rotation basis in five different paddocks.

101 on a rotational basis in five different paddocks.

103 brought into the cold barn on

103 brought into the barn on

104 permanently pastured at the

104 returned to pasture at the

105 with straw and hay bedding and the size of the lying area was

105 with straw bedding and the lying area was

106 could freely choose whether to be in the outdoor walking area

106 could freely choose to be in the outdoor area

110 in the cold barn where

110 in the barn where

110 days together before they were

110 days together, before they were

112 of mineral feed.

112 of minerals. (same again at line 128)

117 The main difference lay in using a McGregor polytunnel, its measurements

117 The main difference lay in using a polytunnel, (supply a brief description of tunnel and the address of McGregor as the supplier) its measurements

119 According to their common practice the sheep were brought into the polytunnel later, only in February

119 According to usual practice on this farm, the sheep were brought into the polytunnel in February

124 the polytunnels only for 75 days

124 the polytunnels for 75 days

142 indoors were visible and were photographed every hour around the clock; 24 photos were gathered per day during the entire survey period

142 indoors were visible and were photographed once every hour during the survey period

149 These included ambient temperature, wind direction (degrees), wind speed (m/s),

149 These included ambient temperature (degrees C), wind direction (compass point??), wind speed (m/s),

158 On Farm B, the survey was carried out for 49 observation days in 2018

158 On Farm B, the observations were carried out for 49 days in 2018

159 were made with the number of animals being

159 were made, with the number of animals being

170 weighted scatterplot smoothing (LOWESS) was used with R function lowess.

170 weighted scatterplot smoothing (LOWESS) was used within R.

194 ewes were observed to be indoors.

194 ewes were observed indoors.

201 Regarding times of day, on both farms ewes spent slightly less time outdoors at night

201 Ewes spent slightly less time outdoors at night on both farms

202 At night, means of 88 and 91% of ewes on the farms chose to stay outdoors

202 On farms A and B respectively, 88 and 91% of ewes chose to stay outdoors at night

203 From 13.00 to 18.00 in Farm A

203 From 13.00 to 18.00 on Farm A

218 fell below -10 °C, a mean of less than 90%

218 fell below -10 °C, on average less than 90%

263 with the high relative

263 with high relative

271 buildings or topology provided

271 buildings or topography provided

273 advantage of the possibility to be outdoors

273 advantage of the opportunity to be outdoors

274 temperatures are well

274 temperatures were well

279 but the proportion of ewes indoors was never more than 50% even at 100% relative humidity.

279 but there was never more than 50% of ewes indoors, even at 100% relative humidity.

289 and it is not concludes that therefore welfare is necessarily improved.

289 and we conclude that welfare is not necessarily improved.

302 an unlikely dominant cause in affecting the narrative of the observations recorded.

302 an unlikely dominant cause in affecting the observations recorded.

310 not be suitable at this period

310 not be suitable during this period

318 a small number of

318 a small proportion of

323 But that they should

323 However, they should

341 Figure S4: Left-hand drawings: I doubt that these were drawn with the left hand, and should more appropriately be given numbers or letters (Figure S4a and S4b 0r Figure S4.1 and Figure S4.2)

There were numerous errors in the Reference section:

Slee, J., Sykes, A.R. Acclimatization

Slee, J., Sykes, A.R. Acclimatisation

Ekesbo, I. Farm Animal Behaviour: characteristic for Assessment of Health and Welfare. CAB International: 369 Cambridge University Press, Cambridge, UK, 2011, 237 pp.

Could not find this reference but found Ekesbo and Gunnarsson 2018

Allan Degen, A., Young, B.A. Effect of air temperature and

Degen, A.A., Young, B.A. Effect of air temperature and

Effect of stocking density on ewes milk yield,

Effect of stocking density on ewes’ milk yield,

Hartung, J. The effect of airbone particulate

Hartung, J. The effect of airborne particulate

Effect of two different

Effects of two different

Mittra, S. 2006. Concepts of organic animal husbandry in organic farming systems. In XIX National Congress of Veterinary Parasitology and National Symposium on national impact of parasitic diseases on livestock health and production, 3–5 February, 2009. Focal theme, changing trends in parasitology: from eggs to genomics 2009, 222–224.

Susamoy, M. (2009). Concepts of organic animal husbandry in organic farming system. In XIX National Congress of Veterinary Parasitology and National Symposium on national impact of parasitic diseases on livestock health and production, 3-5 February, 2009. Focal theme, changing trends in parasitology: from eggs to genomics (pp. 222-224). Department of Veterinary Parasitology, College of Veterinary Science, Guru Angad Dev Veterinary and Animal Sciences University.

Rutter, M. Behaviour of Sheep and Goats. In The Ethology of Domestic Animals; Jensen, P., Ed.; Cab 410 International, Wallingford, UK, 2002; pp. 145–158.

Rutter, S. Mark. "Behaviour of Sheep and Goats 10." The ethology of domestic animals: An introductory text (2002): 145.

Reviewer 2 Report

The effect of Climate Parameters on Sheep Preferences for Outdoors or Indoors at Low Ambient Temperatures

This manuscript details the sheep preferences for access to the outdoors v the indoors in cold climates. Overall, a good manuscript with some improvements to make.

A list of more detailed comments are below.

Title

I would suggest altering the title. Perhaps consider outdoor versus indoor environments in cold climates. The current title seems to repeat itself so another option would be to remove the ‘at low ambient temperatures’ part.

Authors

Just wondering how 4 people can be the Chair of Animal Nutrition? Perhaps this needs another entry for the Chair and one for the other authors located within the Institute of Veterinary Medicine and Animal Science?

General

Consider the use of the term indoor and outdoor. If using them make clear early on what this actually means – access to both indoor and outdoor areas – and that the sheep are not housed only indoors or outdoors e.g. Line 93

Simple summary

Line 17 Remove ‘(when given a choice)’

Line 18-21 Suggest rewording sentence to make findings clearer

Line 22-25 Consider removing these findings as they weren’t in your research, are recommendations from other researchers, and are more suited to your discussion and not your simple summary

Abstract

Line 30 Edit sentence to ‘Sheep preferences for outdoor versus indoor environments, when given free access to both, …’

Line 34 Edit the working to make it clearer that a camera took an image which was analysed. As it reads currently it sounds like a video footage was taken. Add how often the images were taken and over what time period. 24h?

Line 35 Add the % of sheep that preferred to be outdoors rather than just use ‘majority’

Line 40 Suggest editing the final sentence after ‘…lambs and shorn sheep,…’ for ease of reading

Introduction

Line 45 Change to ‘…changes in both conventional and organic farming systems.’

Line 47 change ‘in the winter’ to ‘during this period’

Line 49 Reference for the -20 degrees?

Line 52 Start new sentence with ‘During…

Line 58 change to .’…cold as a stressor, and its subsequent consequences for welfare, decreasing environmental temperatures can reduce feed intake…’

Line 61 Edit to ‘How sheep experience cold and their preferences…’

Line 62 Remove ‘from even a cursory reading of the above paragraph,

Line 63 Change from some time ago to ‘dated (and then give examples by citing older references)

Line 63-65 Remove sentence

Line 65 Change ‘tackling’ to investigating

Line 68 Are ‘A number of comparative studies’ the ones mentioned above in the paragraph or different ones. Needs clarity.

Line 70 Add ‘decreased’ before space allocation + explain ‘worse indoor climate than outdoors’

Line 74 Reference for ventilation and light regimes

Line 80 Start new para with ‘There is a …’ as it doesn’t flow with the piece on organic farming

Line 87 Change to ‘accessing an indoor or outdoor environment at cold temperatures…’

Materials and methods

Line 93 What does ‘kept outside all year’ mean? Grazing outside all year? At pasture? Supplementary fed? Access to shelter?

Line 94 Define walking area. Is this a paddock or more of a standing pad?

Line 95 Explain polytunnel

Line 98-100 Need to very clearly explain how many sheep were used for what trial as this will make the explanation of the analysis numbers easier to comprehend

Line 99 Year of birth of the ewes or was the study done over 2 years

Was there an ethics application completed for this work?

Line 104 Ad’d …permanently returned to pasture at the beginning of May 2018.’

Line 104 Keep consistent on terms – call the uninsulated cold barn either the cold barn or the uninsulated barn  - the manuscript currently swaps between the two terms. Same for outdoor walking area – either outdoor or walking area if not using outdoor walking area

Line 105 Change about to approximately

Line 106 remove ‘by’. Should this be ‘in’? Same with line 123

Line 108 Sheep were shorn

Line 109 ‘The ewe and lamb were placed…’

Line 112-114 Will available feed differences in different areas be considered in the analysis.

Line 118 Were the ewes kept (i.e. housed permanently) in the polytunnel or did they just have free access to it?

Line 119-120 Remove ‘…only in February, i.e.’

Line 120 Replace ‘…as the lambing period’ with ‘which began March 23, 2018

Line 121 November of 2017? Add year

Line 122 Change about to approximately

Line 123 Remove ‘…including the cold days of January and February…’

Line 128 What water was frozen – needs a clearer explanatory sentence

What shelter was in the outdoor walking area?

Line 142 Were all parts of the indoor area visible with the single camera image?

Line 156-160 Needs clearer explanation of how many sheep were used in what part of the study – adding in the average number of ewes makes it more confusing rather than clarifying the trial numbers. Is this the number of ewes seen or the total number of ewes in the trial at that point?

Line 168 Needs explanation of why you have used % not actual number

Results

Line 187 Over what time period were those specific %’s of ewes outside?

Line 192-193 Was this a combination of these two environmental factors or were these different on different days?

Line 194-195 Reword to improve clarity

Is farm A the combination of all the data measured? Given the numbers I am assuming there were no significant differences in this farm between the 2 years but have you checked and reported this – given there were food differences etc.?

Figure 1 Legend doesn’t need ‘staying’ but need to add over what time period

Line 201 ‘slightly less’ - is there a significant difference here or not? Need to clarify

Line 204-205 Change to ‘…Farm B there were no days with fewer than 70%...’

Line 209 should the last work of that title be inside or outside?

Line 245-246 p value here?

Line 250 Change spotted to observed

Line 257 is this the number of observations taken or the percentage of observations? If it’s the percentage then just use percentage ‘Percentage of observations’

Discussion

Line 274-275 Remove ‘…even at temperatures as low as -20..’ repeat

Line 278 Change story to pattern

Line 279 Change choosing to preferring

Line 282 which outdoor areas had no shelter from the prevailing wind direction and how often did the wind blow from that direction?

Line 283 Remove Regarding and just start sentence with ‘Diurnally the ewes were more likely…

Line 289 Need to be clear here – Given we did not measure any specific welfare parameters we cannot conclude that welfare is necessary improved.  However, you can make a statement about the importance of control and choice from any number of behavioral preference/choice experiments e.g. Dawkins

Line 291 Change to ‘increased risk of predation’

Line 292 change to ‘…this study a flock of breeding rams were predated upon in paddocks close to the homestead’. But how relevant is this? What impact does/doesn’t it have on your outcomes?

Line 296 Can you state this? What parameters of cold stress did you measure?

Line 299-315 seems to lack cohesion – suggest making this the part of the discussion where you discuss future research potential

Line 299 Might need to look further into this as dominant sheep can (and usually are) different to leaders in a flock. You were investigating leaders here rather than dominance

Line 303-306 Suggest moving this to methods

Furhter work on lambing is good as would be investigating sex differences, breed differences, other age differences, looking at individual movements within the flock and looking for consistency/differences in individual patterns etc.

Line 327 Add ‘…the local Estonian breeds…’

Reviewer 3 Report

See attached comments
